

# A low-cost monitor for measurement of fine particulate matter and aerosol optical depth. Part 1: Specifications and testing

Eric A. Wendt[1], Casey W. Quinn[2], Daniel D. Miller-Lionberg[3], Jessica Tryner[1], Christian L'Orange[1], Bonne Ford[4], Azer P. Yalin[1], Jeffery R. Pierce[4], Shantanu Jathar[1], and John Volckens[1,2]

[1]Department of Mechanical Engineering, Colorado State University, Fort Collins, 80523, USA
[2]Department of Environmental and Radiological Health Sciences, Colorado State University, Fort Collins, 80523, USA
[3]Access Sensor Technologies, LLC, Fort Collins, USA, 80523, USA
[4] Department of Atmospheric Science, Colorado State University, Fort Collins, 80523, USA

*Correspondence to*: John Volckens (john.volckens@colostate.edu)

**Abstract.** Globally, fine particulate matter ($PM_{2.5}$) air pollution is a leading contributor to death, disease, and environmental degradation. Satellite-based measurements of aerosol optical depth (AOD) are used to estimate $PM_{2.5}$ concentrations across the world, but the relationship between satellite-estimated AOD and ground-level $PM_{2.5}$ is uncertain. Sun photometers measure AOD from the Earth's surface and are often used to improve satellite data; however, reference-grade photometers and $PM_{2.5}$ monitors are expensive and rarely co-located. This work presents the development and validation of the Aerosol Mass and Optical Depth (AMOD) sampler, an inexpensive and compact device that simultaneously measures $PM_{2.5}$ mass and AOD and was designed specifically to be used in citizen science campaigns. The AMOD utilizes a low-cost light-scattering sensor in combination with a gravimetric filter measurement to quantify ground-level $PM_{2.5}$. Aerosol optical depth is measured using optically filtered photodiodes at four discrete wavelengths. Field validation studies revealed agreement within 10% for AOD values measured between co-located AMOD and AErosol RObotics NETwork (AERONET) monitors and for $PM_{2.5}$ mass measured between co-located AMOD and EPA Federal Equivalent Method (FEM) monitors. These results demonstrate that the AMOD can quantify AOD and $PM_{2.5}$ accurately at a fraction of the cost of existing reference monitors.

## 1 Introduction

Fine particulate matter air pollution ($PM_{2.5}$) is a leading contributor to premature death and disease globally (Brauer et al., 2016; Forouzanfar et al., 2016). When inhaled, $PM_{2.5}$ can penetrate deep into the lungs, which can cause long- and short-term health problems (Nel, 2005; Pope and Dockery, 2006). In 2015, approximately 4.2 million premature deaths were attributed to ambient $PM_{2.5}$ exposure (Forouzanfar et al., 2016).

Recently, satellite observations have been used to estimate $PM_{2.5}$ levels at the Earth's surface. These estimates have facilitated global estimates air pollution's impact on public health, especially in remote and resource-limited environments (Brauer et al., 2016). Satellite-based observations provide an estimate of aerosol optical depth (AOD), a dimensionless measure of light extinction in the atmospheric column. Satellite-derived AOD retrievals are then used to estimate $PM_{2.5}$ concentrations



at the Earth's surface (van Donkelaar et al., 2006, 2010; Lv et al., 2016). The relationships between AOD and $PM_{2.5}$ concentration, has been expressed as follows (Snider et al., 2015):

$$PM_{2.5} = \eta \cdot AOD \tag{1}$$

where $\eta$ is a conversion factor between $PM_{2.5}$ and AOD. If $\eta$ is known, satellite AOD estimates can be directly converted to

surface $PM_{2.5}$ concentrations. However, this conversion factor is sensitive to aerosol properties, aerosol composition, surface reflectivity, and vertical profile, all of which can vary across time and space (van Donkelaar et al., 2006, 2010, 2013). Thus, satellite estimates of AOD are prone to error (Boersma and de Vroom, 2006; Brooks, D. R., 2001; Holben et al., 1998; Mims, 1999; Snider et al., 2015).

        To improve satellite AOD retrievals, sun photometers are routinely used to measure AOD from the Earth's surface

(Levy et al., 2005). Sun photometers use photodetectors to measure the incident flux of photons at a given wavelength of light. In conjunction with the Beer-Lambert-Bouger law, aerosol optical depth ($\tau_a$) may be calculated from a Sun photometer measurement per the following equation:

$$\tau_a(\lambda) \;=\; \frac{1}{m}\left(\ln\left(\frac{V_0}{R^2}\right) - ln(V)\right) - \tau_R(\lambda, p) - \tau_{O3} \tag{2}$$

where, $m$ is the relative optical air mass factor, which accounts for different path lengths through the atmosphere when the sun

is at different angles, $R$ is the Earth-sun distance in astronomical units (AU), $V$ is the voltage read by the light detector, $\tau_R$ accounts for Rayleigh scattering by air molecules, $p$ is the pressure, $\lambda$ is the wavelength, $\tau_{O3}$ accounts for ozone absorption, and the extraterrestrial constant, $V_0$, is the voltage produced by incident light at the top of the atmosphere (Brooks, D. R., 2001; Vroom and Amsterdam, 2003). $V_0$ must be evaluated via calibration. The primary method to find $V_0$ is the Langley plot method (Rollin, 2000). By combining the aerosol, ozone absorption, and Rayleigh components into total optical depth ($\tau$) and

rearranging Eq. (2), the following equation (used for a Langley plot) is derived:

$$ln(V) = ln\frac{V_0}{R^2} - \tau \cdot m \tag{3}$$

During a Langley calibration, voltage measurements are taken as the air mass factor changes over the course of a day. The slope of the line gives total optical depth and the intercept at $m = 0$ gives the constant $V_0$. Secondary extraterrestrial constant calibrations may be performed relative to units calibrated via the Langely plot method (Boersma and de Vroom, 2006). Relative

calibrations may be performed by taking coincident measurements with a calibrated and an uncalibrated unit and solving Eq. (2) for $V_0$, with $V$ equal to the light detector voltage from the uncalibrated unit, $\tau_a$ equal to the AOD reported by the calibrated unit, and all other parameters equal to those reported by the uncalibrated unit.

        When AOD is measured at multiple wavelengths, and the Ångström exponent, $\alpha$, is known, AOD for non-measured wavelengths may be inferred from the following relation (Ångström, 1929):

$$\tau_a(\lambda) \;=\; \tau_{a0} \cdot (\lambda_0) \cdot \left(\frac{\lambda}{\lambda_0}\right)^{-\alpha} \tag{4}$$

where $\lambda_0$ is a wavelength measured by the photometer, $\lambda$ is the new wavelength and $\tau_{a0}$ is the measured AOD from the photometer. The Ångström exponent varies depending on the aerosol size distribution; $\alpha$ tends to decrease with increasing



particle size and may not be constant across all wavelength pairs (Eck et al., 1999; O'Neill, 2003). When AOD is measured at multiple wavelengths, curvature in α can be calculated, providing more insight into the aerosol properties (Eck et al., 1999).

Equation (2) assumes that the photometer measures the intensity of monochromatic incident light (Brooks, D. R., 2001). Because the sun is a polychromatic emitter, sun photometers feature light detectors of narrow spectral bandwidth (Shaw, 1983). Light detectors with full-width half-maximum (FWHM) spectral bandwidths of 15 nm or narrower can be approximated as monochromatic (Brooks, D. R., 2001). This requirement precludes the use of inexpensive photodiodes as light detectors because of their wide spectral bandpass (30> nm). The CE318 (Cimel Electronique SAS, Paris, France) used in the Aerosol Robotics Network (AERONET), include photodiodes fitted with optical interference filters to achieve monochromatic detection (Holben et al., 1998). However, high-quality bandpass filters can be cost prohibitive (Holben et al., 1998; Mims, 1999). High cost (e.g., >\$50,000) and maintenance requirements have disqualified the use of expensive interference filter sun photometers in large-scale validation studies and in locations where adequate capital and line power are lacking.

PM$_{2.5}$ samplers co-located with sun photometers can help inform the relationship between AOD and surface PM$_{2.5}$ concentration. The U.S. Environmental Protection Agency, which regulates ambient concentrations of PM$_{2.5}$ mass (Noble et al., 2001), has designated a list of Federal Reference Methods (FRMs) and Federal Equivalent Methods (FEMs) that are used to monitor PM$_{2.5}$ (US EPA, 2017) according to a set of design and performance characteristics (Noble et al., 2001). Like reference-grade sun photometers, the deployment prospects of FRM and FEM monitors are limited by their cost (\$10,000-\$30,000) and the need for line power.

The objective of this work was to develop a user-friendly and low-cost (relative to reference methods) aerosol sampler capable of accurate and precise AOD and PM$_{2.5}$ measurements to be used in citizen science campaigns. We combined filtered-photodiode-based AOD measurements, time-resolved PM$_{2.5}$ measurement via light-scattering, and a time-integrated, gravimetric PM$_{2.5}$ mass measurement to accomplish this objective. The resultant device, the Aerosol Mass and Optical Depth (AMOD) sampler, is capable of simultaneous sun photometry and mass-based particulate matter measurements. In this work, we describe the design of the first-generation AMOD and its validation against reference monitors in real-world environments. In our companion paper (Ford et al., 2019), we describe the pilot citizen science network (Citizen-Enabled Aerosol Measurements for Satellites [CEAMS]) for which the AMOD was designed. We conclude this work by evaluating the shortcomings of this generation of the AMOD and specifying ongoing design improvements, which will be used in future deployments of the CEAMS network.

## 2. Materials and Methods

### 2.1 Instrument Design

The original, wearable UPAS housing was designed to measure personal exposure to aerosols in indoor and work environments (Volckens et al., 2017). Later, UPAS technology was integrated into a weatherproof housing for outdoor deployments to sample wildland fire smoke (Kelleher et al., 2018). The scientific goals of the AMOD development dictated



the UPAS be modified for outdoor and primarily stationary measurement of both PM$_{2.5}$ and AOD. Notable modifications included: a) additional hardware to support AOD measurement capability; b) firmware updates for simultaneous PM$_{2.5}$ and AOD sampling; c) inclusion of a low-cost light-scattering sensor for real-time PM$_{2.5}$ measurement; d) a larger battery and a solar panel for extended battery life; and e) a new weather-resistant housing. A computer-aided rendering highlighting key

internal and structural components of the AMOD is provided in Figure 1.

The design of the AOD measurement system began with the selection of light sensors. Candidate sensors included filtered photodiodes (Murphy et al., 2016), (Intor Inc., Socorro, NM, USA), light emitting diodes (LEDs; Lighthouse LED A-FSMUBC12, WA, USA) (Mims III, 1992), and vertical cavity surface emitting lasers (VCSELs; Vixar Inc. I0-0680M-0000-KP01, Plymouth, MN, USA) – the latter two operated as detectors (Guenter and Tatum, 2002). These sensor options were

evaluated according to cost, variety of available center wavelengths, and spectral bandpass measured at full-width half-maximum (FWHM). Spectral bandpass measurements were made using a tunable light source (Optometrics TLS-25M, Littleton, MA, USA) for LED detectors and a tunable dye laser (Sirah Lasertechnik Allegro, Grevenbroich, Germany) for filtered photodiode and VCSEL detectors (Figure S1). Filtered photodiodes were selected for use in the AMOD due to their sufficiently narrow spectral response bandwidth (<15 nm) and relatively low cost. Filtered photodiodes were also commercially

available at center wavelengths from 400 nm to 1000 nm in increments of approximately 10 nm. No other detector option offered as broad of a selection. LEDs were the least expensive option but were not selected due to their broad spectral response bandwidth. VCSELs were cost prohibitive and exhibited multiple undesirable response peaks (Figure S1).

A printed circuit board containing AOD measurement instrumentation was designed using Autodesk® EAGLE. When populated, this board contained four filtered photodiodes (Figure S2), a quad operational amplifier with low leakage current

(Linear Technology LTC 6242, Milpitas, California, USA) and a 16-bit analog-to-digital converter (Texas Instruments ADS1115, Dallas, Texas, USA). Photodiode wavelengths of 440 nm, 520 nm, 680 nm, and 870 nm were selected to avoid molecular absorption bands, to match wavelengths used by AERONET, and to facilitate aerosol size evaluation (O'Neill, 2003). The board included a solar incidence sensor (Solar MEMS NANO-ISS5, Seville, Spain) and a Wi-Fi module (Espressif Systems ESP8266, Shanghai, China). A GPS (u-blox CAM-M8, Thalwil, Switzerland) provided location data (longitude,

latitude, and altitude) needed to calculate the position of the Sun and estimate ozone optical depth. The AOD measurement board was interfaced with the primary UPAS motherboard via I2C and UART communication. Sampler control firmware was written in C++ on the mbed™ platform (ARM® Ltd., Cambridge, UK).

A light-scattering particulate-matter sensor (Plantower PMS5003, Beijing, China) was integrated into the sampler housing (Figure 1). The PMS5003 included a fan that pulled aerosol through the path of a laser diode and a photodetector.

Particulate matter concentrations were evaluated by a microprocessor embedded in the PMS5003 and accessed via serial communication (Zhou Yong, 2016). Performance of Plantower light scattering sensors has been described previously (AQ-SPEC, n.d.; Kelly et al., 2017).

The AMOD housing was designed using SolidWorks® (ANSYS, Inc., Canonsburg, PA, USA) and built using stereolithographic printing. The housing included four tubes that limited the field of view of the light detectors. Light entered



through 5 mm diameter apertures on the top surface of the housing and subsequently passed through 112 mm long tubes to the active area of the filtered photodiodes. These dimensions yielded an angle of view of 2.56 degrees per sensor, approximately five times the angular diameter of the sun, but within aperture ranges reported for other low-cost sun photometers (Mims, 2002). A narrow viewing angle was required to mitigate errors caused by forward scattered sunlight entering the field of view

of the detector (Torres et al., 2013). The housing also included a sealed inlet and outlet for flow through the PMS5003 sensor. Two sockets with ¼ - 20 Unified National Coarse threads allowed the AMOD to be mounted to standard camera tripods. The housing was weather-resistant when mounted in its intended orientation—with the $PM_{2.5}$ inlet facing the ground and the AOD apertures pointed toward the sun (Figure 2). An O-ring seal prevented leakage through the seam of the housing halves and float-glass windows sealed with foam adhesive protected the optical apertures.

The internal AMOD battery was a 3.6 V, 20.1 Ah custom battery pack comprised of six 18650 lithium ion cells (Panasonic NCR18650B, Kadoma, Japan). The battery was charged via a barrel plug port on the side of the housing. This plug accepted power from a wall charger, external battery, or solar panel (Voltaic® 3.5W) and was watertight when the solar panel cable was attached to the barrel port. The removable solar panel was mounted to the exterior housing using magnets adhered to opposing surfaces on the panel and AMOD housing. Photographs of the external hardware in front and isometric orientations

are provided in Figure 2.

The dimensions of the AMOD were 9.0 cm W x 14.1 cm H x 6.7 cm L and the weight was 0.64 kg. The total cost of goods of the AMOD was less than $1,100 (Table S1). This tabulation was based on a production run of 24 units. The average assembly time for a single AMOD was estimated at two hours, which translated to a cost of $50 at a rate of $25 per hour.

## 2.2 Calibration Procedure

20    One AMOD master unit was calibrated relative to a Cimel CE318 at the DigitalGlobe AERONET site in Longmont, Colorado (Holben et al., 1998). AERONET instruments are calibrated using the Langley plot technique at Mauna Loa observatory—or relative to other AERONET instruments that have been so calibrated—to AOD uncertainties between 0.002 and 0.005 (Eck et al., 1999). The master AMOD calibration consisted of co-located and concurrent measurements taken over the course of two to four hours. The extraterrestrial constant ($V_0$) was determined for each individual measurement by solving

25    Eq. (2) using the AERONET value for AOD. The extraterrestrial constant for the master AMOD unit was then determined by averaging the extraterrestrial constant calculated from each individual measurement. The extraterrestrial constants of all other AMOD units were derived relative to the AMOD master unit by taking a series of simultaneous measurements under variable illumination (Boersma and de Vroom, 2006). The extraterrestrial constant for all other units, $V_{0,i}$, was determined as follows (Boersma and de Vroom, 2006):

30    $$V_{0,i} = V_{0,master} \cdot \rho_i \qquad (5)$$

where $V_{0,master}$ is the extraterrestrial constant of the master unit and $\rho_i$ is the average ratio of photodiode voltage readings from uncalibrated unit $i$ to the master unit.





### 2.3 AOD Calculation Algorithm

We developed AOD calculation firmware using an online, open-source platform (mbed™; ARM® Ltd., Cambridge, UK), which was executed by the on-board microcontroller (STMicroelectronics STM32L152RE, Geneva, Switzerland). Prior to applying Eq. (2) to calculate AOD, the Earth-Sun distance ($R$), the relative optical air mass factor ($m$), and the Rayleigh

5    optical depth ($\tau_R$) were determined in accordance with the measurement location, time, pressure, and temperature. The National Renewable Energy Laboratory (NREL) published a solar position algorithm to calculate azimuth, elevation and zenith angles at uncertainties equal to $\pm 0.0003$ as a function of location, time and for years between 2000 and 6000 (Reda et al., 2008). This algorithm was implemented as a C++ microcontroller code to automate solar calculations for the AMOD. The Earth-Sun distance was calculated directly by the solar position algorithm.

10    The relative optical air mass factor was calculated in terms of the solar zenith angle, $\theta$, as follows (Young, 1994):

$$m = \frac{1.002432 \cdot cos^2(\theta) + 0.148386 \cdot cos(\theta) + 0.0096467}{cos^3(\theta) + 0.149864 \cdot cos^2(\theta) + 0.0102963 \cdot cos(\theta) + 0.000303978} \tag{6}$$

The contributions of Rayleigh scattering and ozone absorption to total optical depth are often substantial and must be subtracted from the total optical depth for accurate AOD measurements (Bodhaine et al., 1999). Rayleigh optical depth is inversely proportional to the fourth power of wavelength, which made accurate quantification especially important for the 440

15    nm and 520 nm channels on the AMOD. Rayleigh optical depth was calculated based on wavelength and ambient pressure measured by an on-board pressure sensor (Bosch Sensortec BMP 280, Kusterdingen, Germany) (Bodhaine et al., 1999). The AMOD's 520 nm and 680 nm channels were within the Chappuis ozone absorption band (450 nm – 850 nm). An empirical model was used to estimate ozone concentrations in Dobson Units (DU)—based on the location and time of the measurement (Van Heuklon, 1979)—which were then used to determine the ozone optical depth (Koontz et al., 2013).

20    Finally, Eq. (2) was applied to determine the total optical depth using sensor inputs; the extraterrestrial constant; and the calculated Earth-Sun distance, relative optical air mass factor, Rayleigh optical depth, and ozone absorption optical depth. AOD, temperature, pressure, relative humidity, time, location, and battery status were then stored on an accessible MicroSD card (Molex 5031821852, Lisle, IL, USA).

### 2.4 User Operation and Measurement Procedure

25    We designed the AMOD to be operated by individuals without a background in aerosol sampling but with an interest in air pollution and citizen science. Care was taken to minimize the complexity of the measurement process. A smartphone application guided the user through a single measurement in a series of steps (Figure S3). Items needed to complete a measurement included an AMOD unit, a filter cartridge loaded with a pre-weighed air-sampling filter, a smartphone (iOS or Android enabled) with the device application ("CEAMS"; available on the Apple App Store and Google Play) downloaded,

30    and a commercial tripod or alternative mount. Prior to initiating a measurement, the operator manually loaded the filter cartridge into position and aligned the AOD sensors with the sun. The alignment process was aided by an integrated pinhole and target apparatus, which was geometrically aligned with the filtered photodiodes (Figures 1, 2). Once the AMOD was



aligned, the operator initiated a sample with the smartphone application. The AMOD then recorded an instantaneous AOD measurement and began sampling air onto the filter under active control of mass flow at 2 L min$^{-1}$. The AMOD also began recording real-time PM$_{2.5}$ levels reported by the PMS5003. Air sampling continued for 48.25 hours before the AMOD automatically shut off. The AMOD maintained a fixed orientation on a tripod for the entire sampling duration—barring any

unintended movements. The AMOD sampled AOD three times over the 48.25-hr sampling period: immediately after the sample started, 24 hours into the sample, and 48 hours into the sample (i.e., at each solar overpass). To partially mitigate errors caused by day-to-day changes in the Sun's position, the AMOD began measuring AOD 15 minutes prior to the 24-hour mark and logged AOD values every 30 seconds until 15 minutes after the 24-hour mark. The operator was able to use this 30-minute window to correct the AMOD's orientation if unintended movements had taken place since the start of the sample. The lowest

AOD values—which corresponded with the highest photodiode signal—from the 30 minute measurement window at 24-hours and 48-hours were taken as the second and third AOD measurements. Upon completion of the sample, the operator downloaded data from the AMOD using the smartphone application and transferred the data to a host server. Measurements taken by citizen scientists using the AMOD are presented in our companion paper, Ford et al., 2019.

## 2.5 Co-location Validation Studies

AMOD AOD measurements were validated in a series of co-location studies using AERONET monitors as the reference method (Holben et al., 1998). AERONET monitors were available at two sites along the Colorado Front Range: NEON-CVALLA (N 40º09'39", W 105º10'01") and Digital Globe (N 40º08'20", W 105º08'13"). Co-location tests took place on three separate days using seven different AMOD units. Between two and four calibrated AMOD units were randomly selected on each testing day and deployed within 50 m of the AERONET monitor. A total of seven AMOD instruments were

used in co-location studies. Four-wavelength AMOD AOD measurements were taken at five-minute intervals over the course of one to four hours on each measurement day. AMOD data were then compared with  Level 1.0 AOD data published in the online AERONET database (Holben et al., 1998). AMOD measurements concurrent within 2 minutes of an AERONET measurement were included in the comparison data set for the wavelength in question. The 500 nm and 675 nm AOD values from the AERONET instruments were adjusted—using Eq. (4) and Ångström coefficients from the AERONET data set—to

match the 520 nm and 680 nm channels on the AMOD, respectively. The 440 nm and 870 nm channels required no adjustment because the AMOD and the AERONET monitors both measure at those wavelengths.

     Time-integrated PM$_{2.5}$ mass concentrations measured using the AMOD filter samples were validated in a series of 48-hr co-location tests conducted with FEM monitors. AMOD units were loaded with 37 mm PTFE filters (MTL PT37P-PF03, Minneapolis, MN USA). The FEM consisted of an EPA-certified Louvered Inlet (PM10 – Mesa Labs SSI2.5, Lakewood, CO

USA) with an inline PM$_{2.5}$ cyclone (URG Corp 2161, Chapel Hill, NC USA) operating at 16.7 L/min. The PM$_{2.5}$ sample was collected on a 47 mm PTFE filter (Tisch Scientific SF18040, North Bend, OH USA). Airflow through the inlet, cyclone, and filter cartridge was maintained by a pump (Gast 86R142-P001B-N270X, Benton Harbor, MI USA) and metered using a mass-flow controller (Alicat MCRW-20SLPM-D/5M, Tucson, AZ USA). Co-location tests occurred in multiple locations—





including downtown Fort Collins, the Colorado State University main campus, and at several personal residences across the city—over a 10-week period. We constructed a custom mount to support the FEM monitors and hold AMOD samplers at 40 cm from the FEM inlet (Figure S4).

The PM$_{2.5}$ mass concentrations measured using the PMS5003 included in the AMOD were evaluated against a co-
located light-scattering FEM monitor (EDM 180, GRIMM, Ainring, Germany) at the Colorado State University main campus (EPA monitoring site 08-069-0009). Light-scattering readings from the AMOD PMS5003 were corrected *post hoc*, relative to the AMOD filter, by multiplying each light-scattering reading by a scaling factor equal to the ratio of the filter measurement to the 48-hr average of the PMS5003.  Hourly averages of the corrected readings were then calculated for comparison to the hourly concentrations reported by the GRIMM EDM 180.

**3. Results and Discussion**

**3.1 AOD Sensor Evaluation**

Close agreement was observed between the AMOD and AERONET monitors for AOD. A comparison plot for all wavelengths and all AERONET co-location testing data is provided in Figure 3 (n = 130 paired measurements for each wavelength). The mean absolute error between the AMOD and AERONET instruments was 0.0079 AOD units (across all
wavelengths), yielding a mean relative error of 10%. These deviations were nearly within the published uncertainties of the AERONET monitors (0.002 – 0.005) (Eck et al., 1999). The mean AOD difference was 0.00063 with 95% confidence upper and lower limits of agreement of 0.026 and -0.024, respectively (Bland and Altman, 1986). A Bland-Altman plot illustrating the mean difference and limits of agreement is provided in Figure S5. The mean difference results indicated a low systematic bias between the two instruments in AOD units. The single set of outlier points shown in Figure 3 was most indicative of a
misalignment error because: 1) the error relative to AERONET was at least 3× the error of all other measurements from the same AMOD unit; 2) measurements taken at the same time and location with different AMOD units exhibited lower error; and 3) the AOD was over-predicted by the AMOD, which is consistent with lower photodiode signal from misalignment. Agreement between AMOD units was comparable to the agreement between AMOD units and AERONET monitors. The average coefficient of variation between AMOD measurements, expressed as a percentage, was 9.0%.

Our evaluation was limited to relatively low AOD values due to the low aerosol concentrations at regional AERONET stations in fall 2017. We do not view this limitation as consequential because the linear dynamic range of the photodetectors used in the AMOD includes AOD values from 0-5 AOD units (specific voltages associated with AOD values are wavelength and calibration dependent). We plan to expand our performance evaluation to a broader range of environmental conditions in future work. Thin cirrus cloud cover on some days likely yielded the highest AOD values; while this was not strictly "aerosol"
optical depth, it allowed for validation across a greater AOD range against the non-cloud-filtered Level 1.0 AERONET data. Compared with AERONET monitors, the main advantages of the AMOD are its low cost and portability. The AMOD (including light-scattering and integrated PM$_{2.5}$ monitoring) has a cost of goods <40× lower than the purchase price of an



AERONET CE318 monitor. The cost of goods—particularly circuit boards and mechanical components—would be reduced at higher quantities. Reference-grade CE318 monitors are advantageous with respect to measurement automation (e.g. sun tracking allows for many measurements throughout the day), the number of AOD wavelengths (nine for the standard model), and the potential for additional sky radiation measurements beyond AOD (Holben et al., 1998).

AERONET co-location results indicate the AMOD can be used to measure AOD with high accuracy when measurements are initiated and overseen by an operator; however, it remains difficult to assess the reliability of unsupervised measurements taken at 24 and 48-hour intervals after the original measurement. Wind and other disturbances can cause slight misalignment to occur between the first and second measurements. Any software adjustments made to compensate for the day-to-day variation in the sun's path assume stability of the AMOD throughout the sampling period. Without automated self-

correction or operator intervention, misalignment manifests itself with erroneously high AOD measures, which are difficult to discriminate from cloud-contaminated measurements. Manual screening requires operator attention, which cannot be expected for a 48+ hour sampling period. Automated cloud screening could benefit from active solar tracking and relatively high frequency measurements (Smirnov et al., 2000). The development of a low-cost solar tracking mount is also the subject of ongoing work. Active tracking would eliminate the need for algorithmic adjustments to account for daily solar position, enable

measurement of daily AOD trends, increase solar power input, and enable robust cloud-screening algorithms. Closed-loop solar tracking will be facilitated by the solar-alignment sensor included in Figure 1. The sensor measures solar alignment based on differential signals between elements of a quadrant photodiode array. Sensor-geometry specific calibration factors enable accurate computation of two-dimensional incidence angles. Incidence angle information will be used in conjunction with a closed-loop motor control algorithm to locate and track the Sun.

AMOD measurements are amenable to re-analysis using ozone data from outside models or retrievals (Wargan et al., 2017). Re-analysis may be used to compensate for $NO_2$ absorption in the 440 nm and 520 nm channels, which is unaccounted for in standard AMOD measurements. We plan to improve ozone compensation calculations as part of the second generation AMOD design. Karavana-Papadimou et al. (2013) modified the model (Van Heuklon, 1979) parameters used in the AMOD algorithm using updated ozone measurements for select European cities (Karavana-Papadimou et al., 2013). The updated

model achieved improved accuracy for European ozone predictions (Karavana-Papadimou et al., 2013). We plan to leverage ozone retrievals across the U.S. to improve the model presently implemented by the AMOD. This approach can be extended into other regions as the need arises.

**3.2 Gravimetric PM$_{2.5}$ Sampler Evaluation**

        Relatively good agreement was found between AMOD gravimetric PM$_{2.5}$ and FEM samplers in the co-location study

(see Figure 4) (Noble et al., 2001). The Pearson correlation between 39 co-located AMOD and FEM measurements was 0.93. The mean absolute error was 0.83 µg m$^{-3}$, corresponding to a mean relative error of 8% between instruments. The mean difference was -0.0037 µg m$^{-3}$ with 95% confidence upper and lower limits of agreement of 1.84 and -1.85 µg m$^{-3}$ respectively (Bland and Altman, 1986). A Bland-Altman plot indicated a low systematic bias between the two instruments as a function of



PM$_{2.5}$ concentration (Figure S6). These results were consistent with the agreement observed in previous work between PM$_{2.5}$ mass concentrations measured using UPAS gravimetric samples and other accepted gravimetric sampling techniques (Arku et al., 2018; Kelleher et al., 2018; Pillarisetti et al., 2019; Volckens et al., 2017). These results are encouraging given the low 48-hour average PM$_{2.5}$ concentrations in Fort Collins during this period (ranging from 3.9 to 12.4 µg m$^{-3}$).

5      Agreement between AMOD units was comparable to the agreement between AMOD units and FEM monitors. The average coefficient of variation between AMOD measurements taken concurrently with different units, expressed as a percentage, was 6.8%. The relative standard deviation for AMOD gravimetric PM$_{2.5}$ measurements collected using duplicate samplers at the same location was 4.9%.

The performance of the AMOD PM$_{2.5}$ sampler was promising in the context of its low cost and compact, portable 10   form factor relative to the FEM. The AMOD cost of goods was less than the purchase price of the FEM used in the co-location studies by a factor of 12. The AMOD was 97% lighter and more compact than the FEM when both were in their stowed configuration. Size comparisons when deployed depend on the apparatus used to mount the AMOD (e.g., camera tripod). The evaluation summarized in Figure 4 was limited to relatively clean conditions in Colorado. In previous works, we have evaluated cyclone performance at concentrations exceeding 20 µg m$^{-3}$ and observed similar agreement with FEM monitors (Kelleher et 15   al., 2018). Further, the UPAS technology (the gravimetric sampling technology with which the AMOD was developed) has been evaluated against reference monitors by several groups at concentrations approaching 1000 µg m-3 with similar results (Arku et al., 2018; Pillarisetti et al., 2019).

**3.3 Light-Scattering PM$_{2.5}$ Sensor Evaluation**

Preliminary co-location results for the AMOD light-scattering sensor indicated relatively good agreement with a 20   GRIMM FEM light-scattering sensor, albeit with an apparent directional bias. A box plot of paired average vs. paired difference PM$_{2.5}$ concentration is provided in Figure 5. Measurement pairs consist of temporally and spatially coincident, hourly average AMOD and FEM PM$_{2.5}$ measurements. Reported AMOD measurements are filter-corrected. Concentrations reported by the FEM ranged from 0 to 17 µg m$^{-3}$. After normalizing the time-resolved AMOD measurements to the filter, the mean absolute error was 1.98 µg m$^{-3}$. The mean difference was 0.04 µg m$^{-3}$ with 95% confidence upper and lower limits of 25   agreement of 5.02 and -4.95 µg m$^{-3}$, respectively (Bland and Altman, 1986). For pair-averaged PM$_{2.5}$ concentrations less than 10 µg m$^{-3}$, AMOD measurements were generally low relative to FEM measurements. For pair-averaged PM$_{2.5}$ concentrations greater than 10 µg m$^{-3}$, AMOD measurements were generally high relative to FEM measurements. This trend held for both corrected and uncorrected AMOD light-scattering sensor measurements (Figure S7).

One limitation associated with the FEM and the PMS5003 is the low digital resolution. Both monitors report integer 30   values (PMS5003 before filter normalization), which can magnify or obscure relative errors at low concentrations. Readings of 0 µg m$^{-3}$ are especially problematic because they cannot be corrected to the filter via scaling factor multiplication. This leaves zero readings uncorrected and tends to magnify the scaling of non-zero readings (Figure 5).



The AMOD light-scattering sensor represents cost savings over reference-quality light-scattering monitors and performance improvements over other low-cost sensors. The cost of goods of the AMOD is 20× less than the purchase prices of two reference quality monitors: the ThermoFisher Tapered Element Oscillating Microbalance (TEOM$^{TM}$) and the GRIMM monitor used in the co-location studies. Filter correction and weatherproof hardware integration may increase the accuracy

and durability of the AMOD light-scattering measurement system compared with stand-alone low-cost sensors.

### 3.4 Wireless Capability

Smartphone connectivity and control is an advantage of the AMOD. The custom AMOD smartphone application serves as a wireless control platform, condensed user manual, and data transfer tool. Wireless control allows the user to start the sampler without the risk of altering an established alignment. Systematic instructions reduce the potential for operator error

and omission. Wireless data transfer is less labor intensive than hardware alternatives (e.g., SD$^{TM}$ card) and can be directly interfaced with a web server via the smartphone Wi-Fi. The present Bluetooth$^{TM}$ smartphone application cannot connect to the AMOD while running, cannot display run data in the app, and downloads data at slow speeds (often in excess of five minutes for a full 48.25-hr dataset). Expanding the web connectivity of the AMOD to include real-time data transfer and visualization using the Wi-Fi chip is the subject of ongoing work. Basic data transfer and real-time visualization capabilities have been

developed for the AMOD using a free Internet of Things (IoT) service (ThingSpeak$^{TM}$) and the ESP8266 Wi-Fi chip. Further development could enable faster data transfer and immediate feedback for participants in AMOD deployments. These capabilities could bolster the scientific potential of AMOD data, provide an interface with other web-connected devices, and facilitate operator engagement.

### 3.5 Potential Sampler Network

The unique combination of AOD, gravimetric filter $PM_{2.5}$, and real-time $PM_{2.5}$ sampling on a compact, user-friendly, and relatively low-cost platform, make the AMOD amenable to large-scale deployment in spatially dense sampling networks. Given these characteristics, the AMOD can deployed in large numbers, by either trained or citizen scientists, to collect spatially dense AOD and $PM_{2.5}$ data sets. These data sets, which can be used to gain a better understanding of spatial and temporal variations in the relationship between AOD and $PM_{2.5}$ concentration, have the potential to improve and expand the use of

satellite AOD-derived estimates of ground-level $PM_{2.5}$ concentrations. We demonstrate the potential to use the AMOD for a citizen science network in our companion paper (Ford et al., 2019), which describes the pilot CEAMS network in northern Colorado.

### Author Contributions

JV, JRP, SJ, and BF designed the study and concept for which the AMOD was designed. EW, CQ, DML, JT, CL, AY, and JV

designed the AMOD device. EW, DML, and JT manufactured prototypes. EW, JT, CL, and AY designed and performed



validation experiments. CQ designed the mobile application. EW led paper with BF, JRP, SJ, and JV; and all co-authors

contributed to interpretation of results and paper editing.

**Acknowledgements**

  This work was supported by the National Aeronautics and Space Administration under Agreement No.

NNX17AF94A and the State of Colorado Office of Economic Development and Information Technology. The authors wish

to thank John Mehaffy, Scott Kelleher, David Brooks, Marilee Long, Lizette van Zyl, Todd Hochwitz (Zebulon Solutions

LLC, Longmont, CO USA), Josh Smith, and Caroline Wendt for their contributions to this work. The authors also thank

Michele Kuester of Digital Globe and Janae Csavina of NEON for their help securing AERONET co-location sites.

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





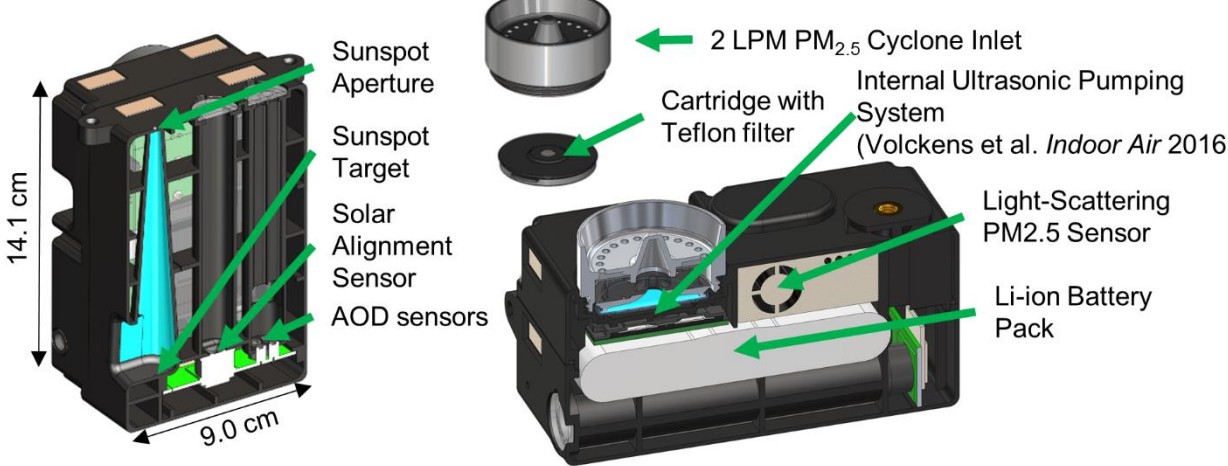

**Figure 1: Computer-aided design rendering of key components of the AMOD including AOD and PM$_{2.5}$ measurement systems, shown as cross-sectional cutaways.**

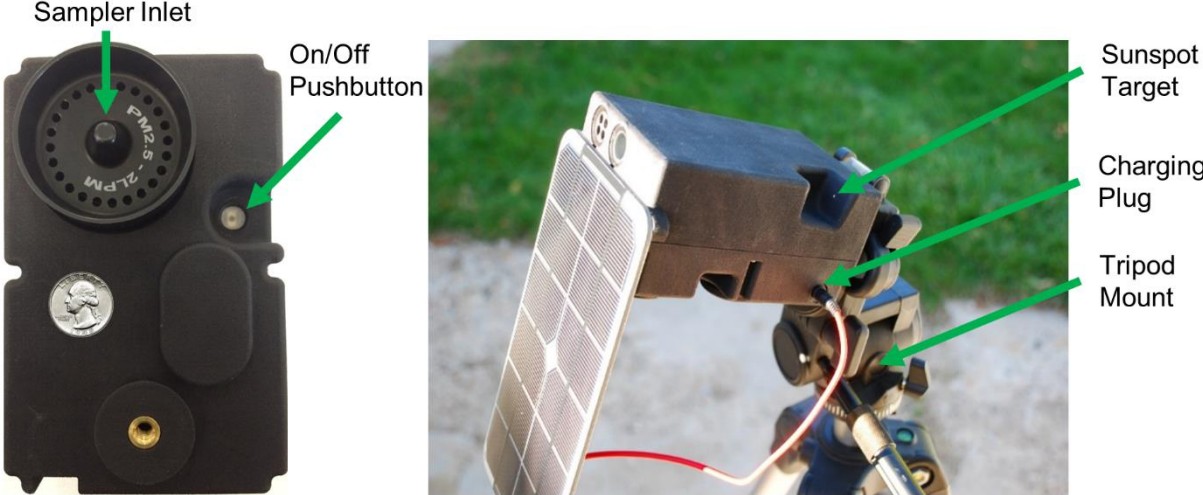

5  **Figure 2: Photographs highlighting AMOD external hardware.**





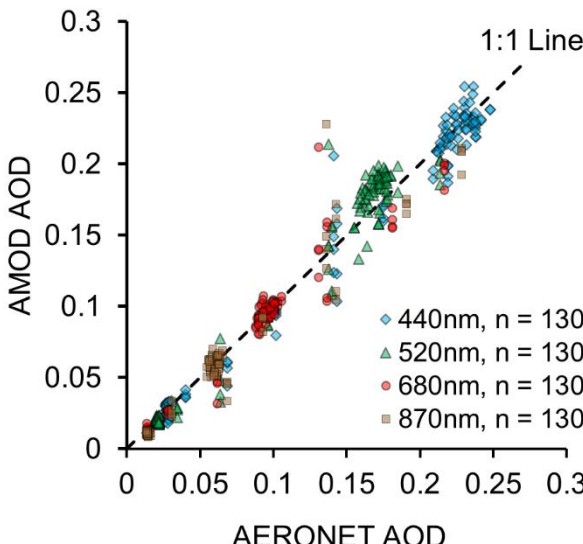

**Figure 3: AERONET vs. AMOD AOD comparison plot. This plot includes all co-located measurements taken across all wavelengths between 3 September and 25 November of 2017.**

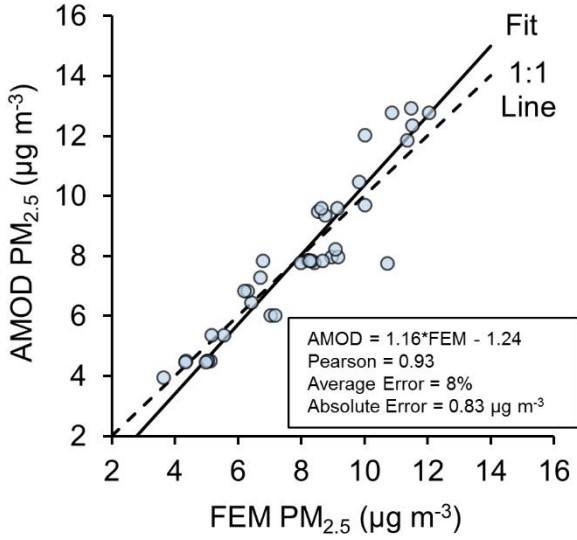

**Figure 4: FEM PM$_{2.5}$ measurements vs. AMOD PM$_{2.5}$ measurements in μg m$^{-3}$ (n = 39). Each data point represents a single 48-hr time-weighted average. All fit statistics were evaluated via Deming regression, assuming equal variance contributions from both measurement devices.**



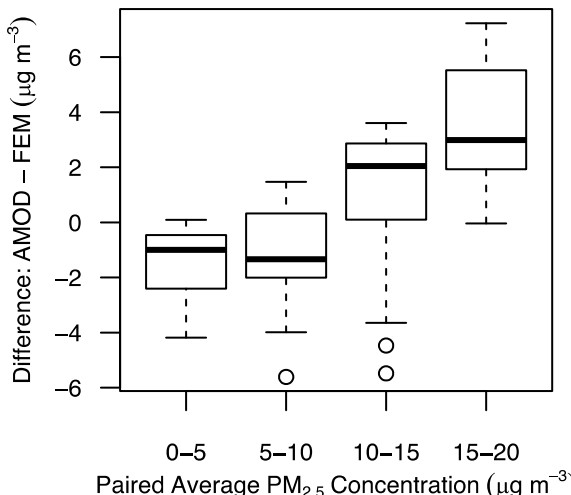

**Figure 5: Binned paired average PM$_{2.5}$ concentration vs. paired difference AMOD and FEM PM$_{2.5}$ measurements. Measurement pairs (n = 96) consist of AMOD and FEM measurements that are temporally and spatially coincident. The four size bins (upper bound inclusive) are 0-5 µg m$^{-3}$ (n = 30), 5-10 µg m$^{-3}$ (n = 24), 10-15 µg m$^{-3}$ (n = 30), and 15-20 µg m$^{-3}$ (n = 12). All light-scattering**
5 **AMOD measurements are corrected to the corresponding AMOD filter measurement.**