# Peer review of "A low-cost monitor for simultaneous measurement of fine particulate matter and aerosol optical depth. Part 1: Specifications and testing"

_Atmospheric Measurement Techniques, 2019_

## Referee Comment (RC1) · Anonymous Referee #1 · 5 May 2019

Wendt et al describe the design and testing of a low-cost monitor that simultaneously measures PM mass and optical depth. The manuscript is topically relevant for AMT and is generally well-written.

I have several comments below, and they generally reflect my opinion that the paper is a bit "light" and would benefit from having certain sections fleshed out in more detail. There are three figures of results (Figures 3-5), and one could argue that Figure 3 is the only one that presents truly new data. As the authors note, the AMOD is an update on the UPAS, so Figure 4 to some extent repeats the validation work for the UPAS. Likewise, several papers cited by the authors, as well as Zamora et al (DOI:

[Figure]

10.1021/acs.est.8b05174), have tested the Plantower sensors, so Figure 5 is not a completely novel result. My comments below reflect places where, in my opinion, the authors could add additional detail and strengthen the paper.

Major comments

(1) Equation 5 assumes that all of the unit-to-unit variability in the photodiodes can be quantified with one voltage, and that all units can be scaled by a single "master" unit. I think that the authors should expand on this discussion and explain how robust of an assumption this is. Even if we assume that all of the manufacturing tolerances are tight (such that manufacturing defects don't contribute to unit-to-unit variability), my overall impression is that many low-cost systems rely on components that can have high unit-to-unit variability. How safe is it to assume that all of that variability can be captured with one parameter?

(2) The long-term robustness and/or drift of the various calibrations, or of the photodiodes themselves, is not discussed. What is a reasonable lifetime for an AMOD? What component is expected to fail first?

(3) AMOD operation relies on the unit remaining still for the entire 48-hr sampling period. How can data be QC'd to make sure that the AMOD didn't move? This is discussed qualitatively on page 9 in the paragraph starting on line 5. However I think it would be much more effective if the authors could show an instance when an AMOD was operated properly and contrast that with an occasion when it was operated improperly and moved. Also, how much movement is tolerable? One can easily imagine the extreme case where someone moves the tripod. But what if the tripod shifts or shakes in the wind? How much does that impact data quality?

(4) Interpretation of the AMOD data seems to implicitly assume that the environment is relatively stable over the 48 hours of measurement - e.g., that PM2.5 concentrations are relatively constant and/or that hours 0, 24, and 48 have similarly sunny conditions. What happens if these conditions are not met? For example, what happens if there

is a large change in PM2.5 concentration over the course of the two days? I could imagine several ways that this could happen, with potentially different impacts on the AOD/PM2.5 relationship. For example: (1) a photochemically active day with high secondary PM could be followed by passage of a weather front or a rain event that dramatically lowers PM2.5, (2) a plume from an industrial source or a wildfire impacts the AMOD site for a portion of the sampling period. Perhaps this means that AMODs are best suited for use outside of urban areas where there are fewer sources.

(5) Figure 4 shows the agreement between the AMOD and FEM PM2.5 measurements. Is the scatter in the data simply a reflection of uncertainty in the AMOD filter measurements? Or are there certain conditions (e.g., meteorology, PM composition on a given day) that lead to better or worse agreement?

(6) How do the authors expect the AMOD to perform in a different environment? My general impression is that the Colorado Front Range is a great place to test the AMODs, since it is often sunny. I'm typing this review in a location where 24 hours ago it was sunny, today the sun is obscured by clouds and there is intermittent rain, and tomorrow will have a mix of clouds and sun. How well do the authors expect their sampling strategy to work in the many parts of the world where day-to-day weather, and even within-day weather, can be extremely variable?

(7) Fig 5 - Does this figure show the raw Plantower output adjusted for the filter measurements, or is some sort of humidity correction also applied?

Minor and Grammatical comments

(1) In equation 3 I assume that tau (with no subscript) is the total optical depth due to aerosol, ozone, and scattering. This is not stated directly in the text. Please clarify.

(2) Page 3, Line 7: The greater than sign seems like it should come before 30.

(3) Page 3, Line 30: UPAS is undefined

---

## Referee Comment (RC2) · Anonymous Referee #2 · 22 Jun 2019

A low-cost monitor for measurement of fine particulate matter and aerosol optical depth. Part 1: Specifications and testing

General Comment: This paper presents the development and validation of low cost sensor to measure PM mass and aerosol optical depth. The paper is interesting and within the scope of AMT. Overall paper is well written but I would recommend some minor changes in manuscript, which are listed below.

1. Introduction section is well written but I feel Line 3-11 at page 3 is bit confusing and need to rewrite for good understanding. 2. Please define what is UPAS (Line 30, page 3)? 3. Both Equation 5 and 6 are very important for this paper approach but there is

lack of understanding in these two equations as well as overall section. I have very similar concern as Reviewer 1, which need to be addressed. 4. Line 2, Page 7: Is mass flow could be changed in your setting? If yes, what would be changes in results? 5. Line 12-13, Page 7: Here need to give some detail of measurements of AMOD taken by citizen scientists. 6. Section 2.5: Please provide proper description of references instruments 7. What is the explanation of performance of AMOD in different atmospheric conditions i.e. rainy, clear sky, high humidity and very low temperature conditions. Assumptions and restrictions of these conditions should be added in the manuscript. 8. I request to add a conclusions section along with scope recommendations

---

## Author Comment (AC1) · 20 Jul 2019

We thank the reviewers for their insightful comments, which have allowed us to produce a stronger manuscript. Our responses to the general and specific comments are given below. Reviewer comments are provided in italics and our responses are given in plain text. Line number references pertain to the revised manuscript with no tracked changes. Quoted text from the revised manuscript is given in blue in this response document. We have provided revised versions of both the manuscript and the SI along with accompanying documents with changes tracked relative to the original submission.

**Reviewer 1:**

Reviewer 1 general comments

*Wendt et al describe the design and testing of a low-cost monitor that simultaneously measures PM mass and optical depth. The manuscript is topically relevant for AMT and is generally well-written.*

*I have several comments below, and they generally reflect my opinion that the paper is a bit "light" and would benefit from having certain sections fleshed out in more detail. There are three figures of results (Figures 3-5), and one could argue that Figure 3 is the only one that presents truly new data. As the authors note, the AMOD is an update on the UPAS, so Figure 4 to some extent repeats the validation work for the UPAS. Likewise, several papers cited by the authors, as well as Zamora et al (DOI: 10.1021/acs.est.8b05174), have tested the Plantower sensors, so Figure 5 is not a completely novel result. My comments below reflect places where, in my opinion, the authors could add additional detail and strengthen the paper.*

**Response:** The reviewer is correct that several studies have evaluated the performance of Plantower sensors (Levy Zamora et al. [2019], DOI: 10.1021/acs.est.8b05174; Bulot et al. [2019, DOI: 10.5281/zenodo.2605402]). The response of these light-scattering sensors to a given PM mass concentration is known to be sensitive to variations in particle size distribution, refractive index, and density. Given that these particle properties vary with time and location, the performance of these sensors is somewhat context specific. We therefore believe that, despite not being the first, our evaluation of the Plantower sensor performance (in addition to the evaluations published by other researchers) represents a valuable addition to the literature. With respect to the UPAS gravimetric sampler, prior evaluations (Volckens et al. [2017], DOI: 10.1111/ina.12318; Arku et al. [2018] DOI: 10.1016/j.envint.2018.02.033; Pillarisetti et al. [2019], DOI: 10.1016/j.envint.2018.11.014]) were conducted at higher concentrations, with different mechanical enclosure designs, and at varying flow rates. We therefore deemed it necessary to include additional evaluations in this manuscript.

Reviewer 1 specific comments

Major comments
1. ***Comment:*** *Equation 5 assumes that all of the unit-to-unit variability in the photodiodes can be quantified with one voltage, and that all units can be scaled by a single "master" unit. I think that the authors should expand on this discussion and explain how robust of an assumption this is. Even if we assume that all of the manufacturing tolerances are tight (such that manufacturing defects don't contribute to unit-to-unit variability), my overall impression is that many low-cost systems rely on components that can have high unit-to-unit variability. How safe is it to assume that all of that variability can be captured with one parameter?*

**Response:** We agree that many "low-cost" sensor instruments do exhibit relatively high unit-to-unit variability in their output, but we did not observe such variability with the filtered photodiodes. Thus, we are confident in the integrity of our transfer calibrations for the measurement of AOD. First, six instruments calibrated via the transfer calibration were independently validated against an AERONET monitor. These validation experiments were performed at a different AERONET site than where the original master calibration took place. Over 80% of the data points depicted in Fig. 3 were from units calibrated via the transfer calibration method. We evaluated the reliability of the transfer calibrations by comparing the performance of the master-calibrated AMOD unit with transfer-calibrated AMOD units. For measurements taken concurrently, we found negligible performance differences between the master unit and transfer-calibrated units. The average difference between transfer-calibrated units and the master unit was 0.006. All transfer-calibrated units measured AOD within 0.01 AOD units of the master unit. Five out of six transfer-calibrated units measured AOD within 0.005 AOD units of the master unit. We have added text to the sections 2.5 and 3.1 to highlight our approach and results in evaluating the transfer calibrations as provided below:

Lines 29-30 page 7 (Methods):
"Device master calibrations were conducted at the Digital Globe site and device validation tests were conducted at NEON-CVALLA."

Lines 30-32 page 7 (Methods):
"Co-location tests took place on three separate days using seven different AMOD units: one calibrated directly relative to AERONET at the Digital Globe site, and six calibrated via the transfer calibration method (Eq. 5)."

Lines 9-12 page 9 (Results):
"We observed negligible performance differences between a master AMOD unit calibrated directly against AERONET instruments and those calibrated via transfer calibrations (Eq. 5). The average difference between units calibrated via the transfer calibration and the master unit was 0.006 AOD units."

Second, both AERONET (Holben et al. 1998, DOI: 10.1016/S0034-4257(98)00031-5) and GLOBE (Brooks and Mims 2001, DOI: 10.1029/2000JD900545) photometers have used transfer calibrations with a similar degree of success.

2. *Comment: The long-term robustness and/or drift of the various calibrations, or of the photodiodes themselves, is not discussed. What is a reasonable lifetime for an AMOD? What component is expected to fail first?*

**Response:** As the AMOD AOD sensor matures, we are gradually gaining insights into the long-term failure modes of the instrument. We expected the calibration of the AOD sensors to fail from the long-term changes of the optical interference filters, based on discussions in prior work (Brooks and Mims [2001], DOI: 10.1029/2000JD900545]; Holben et al. [1998], DOI: 10.1016/S0034-4257(98)00031-5). This turned out to be the case as we prepared for a deployment in China 12 months after the original calibrations took place. We found that two of the four units intended for deployment in China were reporting erroneous AOD data on all four channels, requiring updates to the calibration coefficients. On the remaining units, a single channel was reporting erroneous values. Based on our experience to this point, we recommend updating the AMOD AOD calibrations every six months, which is also the same period used by AERONET. We have added the following text on lines 8-9 on page 6 to state this recommendation:

"We recommend updating the calibration constants of AMOD instruments on a six-month basis to account for changes in optical properties of the filtered photodiodes used here."

We have not yet experienced complete device failures during normal operation. All device failures have come as a result of mechanical damage during handling or calibration. While the AOD sensors do need to be re-calibrated, as stated above, we have not observed any uncorrectable failures of the components. An accurate estimate of device lifetime will require more time to allow failures to manifest.

3. *Comment: AMOD operation relies on the unit remaining still for the entire 48-hr sampling period. How can data be QC'd to make sure that the AMOD didn't move? This is discussed qualitatively on page 9 in the paragraph starting on line 5. However I think it would be much more effective if the authors could show an instance when an AMOD was operated properly and contrast that with an occasion when it was operated improperly and moved. Also, how much movement is tolerable? One can easily imagine the extreme case where someone moves the tripod. But what if the tripod shifts or shakes in the wind? How much does that impact data quality?*

**Response:** For large movements, the on-board accelerometer reports changes in the pitch of the AMOD relative to horizontal. We have added mention to this QC tool on lines 30-34 on page 9 as follows:

"An accelerometer reports the angular pitch of the AMOD relative to horizontal on a 30-second basis. Those data can be used to determine if the AMOD underwent large angular changes (e.g. >2º) relative to the horizontal plane during sample collection. Wind and other disturbances can cause slight misalignment to occur between the first and second measurements that may not be detectable by the accelerometer."

When the deviation from direct sunlight exceeds approximately 0.5º, AMOD photodiodes are no longer uniformly exposed to sunlight, leading to overpredictions of AOD. We have added the following text on lines 27-30 on page 9 to highlight this point as follows:

"The proportions of the AOD apertures permit angular deviations from direct sunlight up to approximately 0.5º for acceptable measurements. In Colorado, for example, the average day-to-day variation—for airmass values less than five—in the solar zenith and azimuth angles is 0.2°. Based on those day-to-day variations, the AMOD is most sensitive to alignment disturbances for measurements taken at the 48-hour mark."

For angular deviations smaller than 5º, future iterations of the AMOD will feature a solar incidence angle sensor based on a quadrant photodiode. This sensor reports the incidence angle of light onto the sensor with a precision of less than 0.1º. We have included an expanded discussion of the QC potential of this sensor in the main text as follows, starting on line 33 on page 9:

"Wind and other disturbances can cause slight misalignment to occur between the first and second measurements that may not be detectable by the accelerometer. A quadrant-photodiode-based solar-alignment sensor, mounted parallel to the AOD sensors, can be used to measure solar incidence angle for deviations smaller than 5º at a precision of 0.1º. The sensor measures solar alignment based on differential signals between elements of a quadrant photodiode array."

In our paper detailing our citizen-science deployment in Northern Colorado of ~20 AMOD's (Discussion paper here: https://www.atmos-meas-tech-discuss.net/amt-2019-109/), we give multiple instances of successful AOD measurements taken over the 48-hour period by a single instrument. Small disturbances that fell within the tolerable range (<0.5º from direct sunlight), still allowed for successful measurements. Given the low margin of error of the instrument alignment, even minor perturbations would result in near-zero photodiode signal, which can be easily separated from successful measurements, particularly on days without cloud cover.

Here is the updated discussion on misalignment in its entirety starting at line 25 on page 9:

"AERONET co-location results indicate that the AMOD can be used to measure AOD with high accuracy when measurements are initiated and overseen by an operator; however, it remains difficult to assess the reliability of unsupervised measurements taken at 24 and 48-hour intervals after the original measurement. The proportions of the AOD apertures permit angular deviations from direct sunlight up to approximately 0.5º for acceptable measurements. In Colorado, for example, the average day-to-day variation in the solar zenith and azimuth angles is 0.2° for airmass values less than five. Based on those day-to-day variations, the AMOD is most sensitive to alignment disturbances for measurements taken at the 48-hour mark. An accelerometer reports the angular pitch of the AMOD relative to horizontal on a 30-second basis. Those data can be used to determine if the AMOD underwent large angular changes (e.g. >2º) relative to the horizontal plane during sample collection. Wind and other disturbances can cause slight misalignment to occur between the first and second measurements that may not be detectable by the accelerometer. To help catch these events, a quadrant-photodiode-based solar-alignment sensor, mounted parallel to the AOD sensors, could be added to the AMOD to measure solar incidence angle for deviations smaller than 5º at a precision of 0.1º. The sensor measures solar alignment based on differential signals between elements of a quadrant photodiode array. Without automated self-correction or operator intervention, misalignment manifests itself with erroneously high AOD measures, which are similar to cloud-contaminated measurements. Manual screening requires operator attention, which cannot be expected for a 48+ hour sampling period; however, erroneously high AOD measures, due to either misalignment or cloud contamination, can be identified and eliminated using an automated data screening algorithm.

The development of a low-cost solar tracking mount is also the subject of ongoing work. Active tracking would eliminate the need for algorithmic adjustments to account for daily solar position, enable measurement of daily AOD trends, increase solar power input, and enable robust cloud-screening algorithms. Closed-loop solar tracking will be facilitated by a quadrant diode solar-alignment sensor. Sensor-geometry specific calibration factors enable accurate computation of two-dimensional incidence angles. Incidence angle information will be used in conjunction with a closed-loop motor control algorithm to locate and track the Sun."

As mentioned in the manuscript, AOD misalignment was one of our chief concerns with the manual version of the AMOD. Learning from the quality-control difficulties we encountered with our first deployment, we are planning to include a closed-loop motorized solar tracking feature on the next iteration of the AMOD instrument.

4. **Comment:** *Interpretation of the AMOD data seems to implicitly assume that the environment is relatively stable over the 48 hours of measurement - e.g., that PM2.5 concentrations are relatively constant and/or that hours 0, 24, and 48 have similarly sunny conditions. What*

*happens if these conditions are not met? For example, what happens if there is a large change in PM2.5 concentration over the course of the two days? I could imagine several ways that this could happen, with potentially different impacts on the AOD/PM2.5 relationship. For example: (1) a photochemically active day with high secondary PM could be followed by passage of a weather front or a rain event that dramatically lowers PM2.5, (2) a plume from an industrial source or a wildfire impacts the AMOD site for a portion of the sampling period. Perhaps this means that AMODs are best suited for use outside of urban areas where there are fewer sources.*

**Response:** We have included the plantower sensor to detect changes in $PM_{2.5}$ during the 48-hour sample. Additionally, to date, most studies of $PM_{2.5}$ exposure and health use daily (i.e., responses to acute exposure) or annual (i.e., responses to chronic exposure) mean $PM_{2.5}$ concentrations for exposure. "Satellite-based $PM_{2.5}$" estimates (e.g., van Donkelaar et al., 2006; 2010) use ~mid-day AOD from satellites to gain information on the daily mean $PM_{2.5}$ concentrations (and these daily mean values may be averaged to annual values). These satellite-based estimates suffer from similar issues in that the AOD/$PM_{2.5}$ relationship may change dramatically during the time of day, and mid-day conditions may not represent daily mean conditions (this is discussed in our Part 2 discussion paper: https://www.atmos-meas-tech-discuss.net/amt-2019-109/). Certainly sampling $PM_{2.5}$ over two days rather than one makes these issues worse, but all of the potential issues raised by the reviewer above hold for prior satellite-based $PM_{2.5}$ studies. With the Plantower sensor, we can investigate $PM_{2.5}$ variability within 2-day and 1-day periods, allowing us to investigate the use of single-time AOD measurements as a proxy for time-averaged $PM_{2.5}$ concentrations.

5. *Comment: Figure 4 shows the agreement between the AMOD and FEM PM2.5 measurements. Is the scatter in the data simply a reflection of uncertainty in the AMOD filter measurements? Or are there certain conditions (e.g., meteorology, PM composition on a given day) that lead to better or worse agreement?*

**Response:** We did not observe systematic changes in the performance of the AMOD filter measurements in response to changes in conditions. Given that these performance results are similar to those we have observed with previous iterations of our filter-based sampler, we are confident that the scatter in the data is a reflection of the uncertainty of the measurement. We also note that the FEM instrument is also subject to measurement error and imprecision, so some of the observed scatter may reflect uncertainty in the FEM filter instrument, too.

6. *Comment: How do the authors expect the AMOD to perform in a different environment? My general impression is that the Colorado Front Range is a great place to test the AMODs, since it is often sunny. I'm typing this review in a location where 24 hours ago it was sunny, today the sun is obscured by clouds and there is intermittent rain, and tomorrow will have a mix of clouds*

*and sun. How well do the authors expect their sampling strategy to work in the many parts of the world where day-to-day weather, and even within-day weather, can be extremely variable?*

**Response:** We have confidence in the ability of the AMOD $PM_{2.5}$ filter and Plantower sensor to perform well in environments outside of typical Colorado Front Range weather. The $PM_{2.5}$ sampler component of the AMOD has been tested at higher concentrations. Kelleher et al. (2018) (https://doi.org/10.5194/amt-11-1087-2018) field-tested the $PM_{2.5}$ component at concentrations exceeding 20 µg m$^{-3}$. Further, the UPAS technology (the gravimetric sampling technology with which the AMOD was developed) has been evaluated against reference monitors by several groups at concentrations approaching 1000 µg m$^{-3}$ with similar results to reference instruments (Volckens et al. [2017] [https://doi.org/10.1111/ina.12318], Arku et al. [2018] [https://doi.org/10.1016/j.envint.2018.02.033], Pillarisetti et al. [2019] [https://doi.org/10.1016/j.envint.2018.11.014]). In particular, Arku et al. used the UPAS to reliably measure $PM_{2.5}$ in 10 countries including Bangladesh, Brazil, Chile, China, Colombia, India, Pakistan, South Africa, Tanzania, and Zimbabwe. Additionally, while these studies have not shown any issue with the filter loading, the devices can be modified to run for different time durations or sample at a different rate if there are concerns about filter loading. The following text on lines 6-11 on page 11, modified slightly from the original submission for clarity and completeness, highlights this point:

"The evaluation summarized in Figure 4 was limited to relatively clean conditions in Colorado. In previous works, we have evaluated cyclone performance at concentrations from 15 µg m$^{-3}$ to 40 µg m$^{-3}$ and observed similar agreement with FEM monitors (Kelleher et al., 2018). Further, the UPAS technology (the gravimetric sampling technology with which the AMOD was developed) has been evaluated against reference monitors at concentrations approaching 1000 µg m$^{-3}$ and in over 10 different countries with similar results (Arku et al., 2018; Pillarisetti et al., 2019)."

The AMOD AOD sensor is less mature than its $PM_{2.5}$ counterparts and has been evaluated, to date, under relatively limited conditions. However, we do not believe this limitation is consequential, given the high dynamic range of the photodiode light detectors used here. We also show consistent results across a fairly large AOD range during our testing under thin cirrus. In other words, we have found that these detectors are linear across several orders of magnitude of incident light intensity; thus, the sensors used in the AMOD should respond to AOD values up to ~5. Below are some example measurements from the AMOD in China showing measurements at high AOD. Note that the AERONET monitors included on the plot were not co-located with the AMODs.

[Figure]

With respect to weather, the AOD sensors rely upon clear skies to measure AOD correctly. This is a limitation of all AOD-sensing instruments, including those on satellites, which like the AMOD, typically measure AOD in a given location one time per day. From a mechanical and electrical weatherproofing perspective, the AMOD mechanical housing is robust to rain and snow. Therefore, the AMOD will continue sampling $PM_{2.5}$ and attempting AOD measurements under variable weather and will measure AOD correctly when the sun is once again detectable. In locations with higher winds, we would need to consider more stable mounting options than low-cost tripods to ensure weather does not adversely affect measurements.

7. **Comment:** *Fig 5 - Does this figure show the raw Plantower output adjusted for the filter measurements, or is some sort of humidity correction also applied?*

**Response:** Figure 5 shows the Plantower output (with Plantower's proprietary atmospheric correction) adjusted for the filter measurement. We did not apply any humidity correction because the ambient humidity was consistently under 50% throughout the measurement period; thus, humidity artifacts are likely to be negligible for the data collected here.

Reviewer 1 minor/grammatical comments

8. **Comment:** *In equation 3 I assume that tau (with no subscript) is the total optical depth due to aerosol, ozone, and scattering. This is not stated directly in the text. Please clarify.*

**Response:** The reviewer is correct. This is stated in lines 18-20 on page 2 prior to the equation in the text as follows:

"By combining the aerosol, ozone absorption, and Rayleigh components into total optical depth ($\tau$) and rearranging Eq. (2), the following equation (used for a Langley plot) is derived:"

9. **Comment:** *Page 3, Line 7: The greater than sign seems like it should come before 30.*

**Response:** We have corrected this mistake and lines 6-7 on page 3 now read:

"This requirement precludes the use of inexpensive photodiodes as light detectors because of their wide spectral bandpass (>30 nm)."

10. **Comment:** *Page 3, Line 30: UPAS is undefined*

**Response:** Thank you for pointing out this mistake. UPAS stands for Ultrasonic Personal Aerosol Sampler. We have added lines 3-5 on page 4 to define UPAS as follows:

"The AMOD design was based on a low-cost gravimetric sampler known as the Ultrasonic Personal Aerosol Sampler (UPAS), which was developed through prior work (Volckens et al., 2017)."

**Response to Reviewer #2**

Reviewer 2 general comments

This paper presents the development and validation of low cost sensor to measure PM mass and aerosol optical depth. The paper is interesting and within the scope of AMT. Overall paper is well written but I would recommend some minor changes in manuscript, which are listed below.

Reviewer 2 specific comments

1. ***Comment:*** *Introduction section is well written but I feel Line 3-11 at page 3 is a bit confusing and need to rewrite for good understanding.*

**Response:** We have updated the section for additional clarity as follows on lines 3-16 on page 3:

"Equation (2) assumes that the photometer measures the intensity of monochromatic light (Brooks, D. R., 2001). Because the sun emits polychromatic light, sun photometers feature light detectors with narrow spectral bandwidth (Shaw, 1983). Light detectors with full-width half-maximum (FWHM) spectral bandwidths of 15 nm or narrower can be approximated as monochromatic, permitting the application of Eq. 2 with negligible error (Brooks, D. R., 2001). The requirement of approximately monochromatic detection precludes the use of photodiode sensors with broad spectral bandpass (>30 nm). CE318 (Cimel Electronique SAS, Paris, France) sun photometers used in the Aerosol Robotics Network (AERONET), a global reference network of sun photometers, include photodiodes fitted with optical interference filters to achieve approximately monochromatic detection (Holben et al., 1998). However, high-quality bandpass filters can be cost prohibitive (e.g. >\$100) (Holben et al., 1998; Mims, 1999). The high cost of the light-sensing elements partially contributes to the overall high cost (e.g. >\$50,000) of sun photometers used in AERONET. Previous studies have used Light Emitting Diodes (LEDs) acting as detectors as a low-cost alternative to optical interference filters (Boersma and de Vroom, 2006; Brooks, D. R., 2001; Mims III, 1992). Other studies have used relatively low-cost (<\$30) integrated optical filter and photodiode modules (Murphy et al., 2016). The increasing availability of inexpensive alternatives has facilitated the production of relatively inexpensive sun photometers, which are more cost-effective for large-scale deployments (Brooks, D. R., 2001)."

2. ***Comment:*** *Please define what is UPAS (Line 30, page 3)?*

**Response:** Thank you for pointing out this mistake. UPAS stands for Ultrasonic Personal Aerosol Sampler. We have added lines 3-5 on page 4 to define UPAS as follows:

"The AMOD design was based on a low-cost gravimetric sampler known as the Ultrasonic Personal Aerosol Sampler (UPAS), which was developed through prior work (Volckens et al., 2017)."

3. **Comment:** *Both Equation 5 and 6 are very important for this paper approach but there is lack of understanding in these two equations as well as overall section. I have very similar concern as Reviewer 1, which need to be addressed.*

**Response:** We agree that many "low-cost" sensor instruments do exhibit relatively high unit-to-unit variability in their output, but we did not observe such variability with the filtered photodiodes. Thus, we are confident in the integrity of our transfer calibrations for the measurement of AOD. First, six instruments calibrated via the transfer calibration were independently validated against an AERONET monitor. These validation experiments were performed at a different AERONET site than where the original master calibration took place. Over 80% of the data points depicted in Fig. 3 were from units calibrated via the transfer calibration method. We evaluated the reliability of the transfer calibrations by comparing the performance of the master-calibrated AMOD unit with transfer-calibrated AMOD units. For measurements taken concurrently, we found negligible performance differences between the master unit and transfer-calibrated units. The average difference between transfer-calibrated units and the master unit was 0.006. All transfer-calibrated units measured AOD within 0.01 AOD units of the master unit. Five out of six transfer-calibrated units measured AOD within 0.005 AOD units of the master unit. We have added text to the sections 2.5 and 3.1 to highlight our approach and results in evaluating the transfer calibrations as provided below:

Lines 29-30 page 7 (Methods):
"Device master calibrations were conducted at the Digital Globe site and device validation tests were conducted at NEON-CVALLA."

Lines 30-32 page 7 (Methods):
"Co-location tests took place on three separate days using seven different AMOD units: one calibrated directly relative to AERONET at the Digital Globe site, and six calibrated via the transfer calibration method (Eq. 5)."

Lines 9-12 page 9 (Results):
"We observed negligible performance differences between a master AMOD unit calibrated directly against AERONET instruments and those calibrated via transfer calibrations (Eq. 5). The average difference between units calibrated via the transfer calibration and the master unit was 0.006 AOD units."

Second, both AERONET (Holben et al. 1998, DOI: 10.1016/S0034-4257(98)00031-5) and GLOBE (Brooks and Mims 2001, DOI: 10.1029/2000JD900545) photometers have used transfer calibrations with a similar degree of success.

4. **Comment:** *Line 2, Page 7: Is mass flow could be changed in your setting? If yes, what would be changes in results?*

**Response:** Yes, our pumping hardware and software are capable of operating at different flow rates. However, we specifically designed the cyclone used for AMOD validation testing and our early deployments for operation at 2 L min$^{-1}$. If the flow rate was configured differently without replacing the cyclone, the collected sample would not accurately represent the ambient PM$_{2.5}$ concentration: lower flow rates would cause undersampling errors and higher flow rates would cause oversampling errors. We designed the AMOD housing, the cyclone body, and the AMOD-cyclone mechanical interface such that cyclones can easily be replaced for different flow rate selections. For example, in phase two of our citizen-science deployment, we plan to run the sampler at 1 L min$^{-1}$, using a specially designed cyclone and appropriately modified configuration software. We have already conducted multiple successful field tests at 1 L min$^{-1}$ with multiple AMOD units.

5. **Comment:** *Line 12-13, Page 7: Here need to give some detail of measurements of AMOD taken by citizen scientists.*

**Response:** We have submitted this manuscript as part 1 of a two-part work. Part 1 details the device design and validation, and Part 2 details the results from our citizen science pilot campaign in Northern Colorado. Here is a link to the AMT discussion paper for Part 2 of our work: https://www.atmos-meas-tech-discuss.net/amt-2019-109/.

6. **Comment:** *Section 2.5: Please provide proper description of references instruments.*

**Response:** We have added additional detail to the descriptions of the AOD and real-time PM$_{2.5}$ reference monitors. The modified paragraph for AOD testing starting on line 23 on page 7 as follows:

"AMOD AOD measurements were validated in a series of co-location studies using AERONET CE318 monitors as the reference method (Holben et al., 1998). CE318 monitors used in the co-location studies had a 1.2º full angle field of view and measured AOD at eight wavelengths: 340 nm, 380 nm, 440 nm, 500 nm, 675 nm, 870 nm, 1020 nm, and 1640 nm (Holben et al., 1998). The CE318 monitors used stepping motors and closed loop control to locate and track the sun and reported measurements every 3-15 minutes when solar alignment was achieved (Holben et al., 1998). AERONET monitors were available at two sites along the Colorado Front Range:

NEON-CVALLA (N 40º09'39", W 105º10'01") and Digital Globe (N 40º08'20", W 105º08'13"). Device master calibrations were conducted at the Digital Globe site and device validation tests were conducted at NEON-CVALLA. Co-location tests took place on three separate days using seven different AMOD units: one calibrated directly relative to AERONET at the Digital Globe site, and six calibrated via the transfer calibration method (Eq. 5). Between two and four calibrated AMOD units were randomly selected on each testing day and deployed within 50 m of the AERONET monitor. Four-wavelength AMOD AOD measurements were taken at five-minute intervals over the course of one to four hours on each measurement day. AMOD data were then compared with Level 1.0 AOD data published in the online AERONET database (Holben et al., 1998). AMOD measurements concurrent within 2 minutes of an AERONET measurement were included in the comparison data set for the wavelength in question. The 500 nm and 675 nm AOD values from the AERONET instruments were adjusted—using Eq. (4) and Ångström coefficients from the AERONET data set—to match the 520 nm and 680 nm channels on the AMOD, respectively. The 440 nm and 870 nm channels required no adjustment because the AMOD and the AERONET monitors both measure at those wavelengths."

The modified paragraph for real-time PM$_{2.5}$ testing on lines 18-26 on page 8 is provided below:

"The PM$_{2.5}$ mass concentrations measured using the PMS5003 included in the AMOD were evaluated against a co-located light-scattering FEM monitor (GRIMM EDM 180, Ainring, Germany) at the Colorado State University main campus (EPA monitoring site 08-069-0009). The GRIMM utilized a 660 nm diode laser cell couple with a light detector to measure particle concentrations based on light scattering. Flow through the GRIMM was maintained at 1.2 L min$^{-1}$. PM$_{2.5}$ readings from the AMOD PMS5003 were corrected *post hoc*, relative to the AMOD filter, by multiplying each light-scattering reading by a scaling factor equal to the ratio of the filter measurement to the 48-hr average of the PMS5003. The PMS5003 outputs uncorrected PM-$_{2.5}$ concentrations as well as PM--$_{2.5}$ concentrations with a proprietary correction factor for use under atmospheric conditions. We used the corrected data output by the PMS5003 for our analyses. Hourly averages of the corrected readings were then calculated for comparison to the hourly concentrations reported by the GRIMM EDM 180."

7. ***Comment:*** *What is the explanation of performance of AMOD in different atmospheric conditions i.e. rainy, clear sky, high humidity and very low temperature conditions. Assumptions and restrictions of these conditions should be added in the manuscript.*

**Response:** We have confidence in the ability of the AMOD PM$_{2.5}$ filter and Plantower sensor to perform well in environments outside of typical Colorado Front Range weather. The PM$_{2.5}$ sampler component of the AMOD has been tested at higher concentrations. Kelleher et al. (2018) (https://doi.org/10.5194/amt-11-1087-2018) field-tested the PM$_{2.5}$ component at concentrations

exceeding 20 µg m$^{-3}$. Further, the UPAS technology (the gravimetric sampling technology with which the AMOD was developed) has been evaluated against reference monitors by several groups at concentrations approaching 1000 µg m$^{-3}$ with similar results to reference instruments (Volckens et al. [2017] [https://doi.org/10.1111/ina.12318], Arku et al. [2018] [https://doi.org/10.1016/j.envint.2018.02.033], Pillarisetti et al. [2019] [https://doi.org/10.1016/j.envint.2018.11.014]). In particular, Arku et al. used the UPAS to reliably measure PM$_{2.5}$ in 10 countries including Bangladesh, Brazil, Chile, China, Colombia, India, Pakistan, South Africa, Tanzania, and Zimbabwe. Additionally, while these studies have not shown any issue with the filter loading, the devices can be modified to run for different time durations or sample at a different rate if there are concerns about filter loading. The following text on lines 6-11 on page 11, modified slightly from the original submission for clarity and completeness, highlights this point:

"The evaluation summarized in Figure 4 was limited to relatively clean conditions in Colorado. In previous works, we have evaluated cyclone performance at concentrations from 15 µg m$^{-3}$ to 40 µg m$^{-3}$ and observed similar agreement with FEM monitors (Kelleher et al., 2018). Further, the UPAS technology (the gravimetric sampling technology with which the AMOD was developed) has been evaluated against reference monitors at concentrations approaching 1000 µg m$^{-3}$ and in over 10 different countries with similar results (Arku et al., 2018; Pillarisetti et al., 2019)."

The AMOD AOD sensor is less mature than its PM$_{2.5}$ counterparts and has been evaluated, to date, under relatively limited conditions. However, we do not believe this limitation is consequential, given the high dynamic range of the photodiode light detectors used here. We also show consistent results across a fairly large AOD range during our testing under thin cirrus. In other words, we have found that these detectors are linear across several orders of magnitude of incident light intensity; thus, the sensors used in the AMOD should respond to AOD values up to ~5. Below are some example measurements from the AMOD in China showing measurements at high AOD. Note that the AERONET monitors included on the plot were not co-located with the AMODs.

[Figure]

With respect to weather, the AOD sensors rely upon clear skies to measure AOD correctly. This is a limitation of all AOD-sensing instruments, including those on satellites, which like the AMOD, typically measure AOD in a given location one time per day. From a mechanical and electrical weatherproofing perspective, the AMOD mechanical housing is robust to rain and snow. Therefore, the AMOD will continue sampling $PM_{2.5}$ and attempting AOD measurements under variable weather and will measure AOD correctly when the sun is once again detectable. In locations with higher winds, we would need to consider more stable mounting options than low-cost tripods to ensure weather does not adversely affect measurements.

8. ***Comment: I request to add a conclusions section along with scope recommendations.***

**Response:** We have added a conclusion and scope recommendations section on lines 22-29 on page 12 as follows:

"The AMOD is a lightweight and compact alternative to the instruments typically used to sample AOD and $PM_{2.5}$. The AMOD represents a substantial cost saving compared with alternative AOD and $PM_{2.5}$ mass concentration sampling equipment. In field testing, the AMOD exhibit agreement within 10% when compared with AOD and $PM_{2.5}$ reference instruments. The AMOD has been validated only in a relatively clean air in Colorado in fall and wintertime; more validation in other environments of varying pollution/weather patterns is needed. The small size, durability, increased sampling capabilities and relatively low cost of the AMOD make it a viable option for large scale and spatially dense deployments. Such data sets have the potential to

facilitate the calibration and validation of satellite-based sensors as they progress toward higher spatial resolution measurement capabilities."

Below is the text specifically related to scope recommendations on lines 25-26 on page 12:

"The AMOD has been validated only in a relatively clean airshed in Colorado in wintertime; more validation in other environments of varying pollution/weather patterns is needed."